# BESA: Pruning Large Language Models with Blockwise Parameter-Efficient Sparsity Allocation

**Peng Xu**[1,2]   **Wenqi Shao**[*2]   **Mengzhao Chen**[2]   **Shitao Tang**

**Kaipeng Zhang**[2]   **Peng Gao**[2]   **Fengwei An**[3]   **Yu Qiao**[2]   **Ping Luo**[*1,2]

[1] The University of Hong Kong   [2] OpenGVLab, Shanghai AI Laboratory

[3] Southern University of Science and Technology

## Abstract

Large language models (LLMs) have demonstrated outstanding performance in various tasks, such as text summarization, text question-answering, and etc. While their performance is impressive, the computational footprint due to their vast number of parameters can be prohibitive. Existing solutions such as SparseGPT and Wanda attempt to alleviate this issue through weight pruning. However, their layer-wise approach results in significant perturbation to the model's output and requires meticulous hyperparameter tuning, such as the pruning rate, which can adversely affect overall model performance. To address this, this paper introduces a novel LLM pruning technique dubbed blockwise parameter-efficient sparsity allocation (BESA) by applying a blockwise reconstruction loss. In contrast to the typical layer-wise pruning techniques, BESA is characterized by two distinctive attributes: i) it targets the overall pruning error with respect to individual transformer blocks, and ii) it allocates layer-specific sparsity in a differentiable manner, both of which ensure reduced performance degradation after pruning. Our experiments show that BESA achieves state-of-the-art performance, efficiently pruning LLMs like LLaMA1, and LLaMA2 with 7B to 70B parameters on a single A100 GPU in just five hours. Code is available at here.

## 1 Introduction

Large language models (LLMs) have demonstrated remarkable performance in a wide range of NLP tasks, including language modeling, code generation, machine translation, sentiment analysis, and question answering (Zhang et al., 2022a; Touvron et al., 2023a;b; Xu et al., 2023; Team, 2023; Zeng et al., 2022). However, LLMs have a vast number of parameters, resulting in high memory consumption and slow inference speed (Dettmers et al., 2022). For example, it requires 335GB GPU memory (*i.e.* five A100 GPU with 80G memory) to load its parameters in FP16 of Falcon-180B (Penedo et al., 2023), which corresponds to the inference speed of merely 4 tokens per second. Thus, there has been considerable interest in compressing LLMs to make LLMs more efficient and practical for deployment in various applications.

One of the approaches to compress a network is weight pruning. Although it has a long history in model compression (Hassibi et al., 1993; Hassibi & Stork, 1992), few pieces of work can be used to prune LLMs due to the requirement of extensive retraining. Recent studies, such as SparseGPT (Frantar & Alistarh, 2023) and Wanda (Sun et al., 2023) aim to tackle this challenge by reconstructing the layer-wise output of LLMs, as illustrated in Fig.1(c). Specifically, SparseGPT proposes to prune unimportant with an importance metric derived from the hessian matrix. and then reconstruct layer-wise output. Moreover, Wanda removes intricate computation in SparseGPT by only leveraging the product of weight and activation magnitudes.

While these approaches can eliminate considerable unnecessary weights, they typically operate within each weight by minimizing each layer's pruning error, which has two drawbacks. First, layer-wise pruning error minimization does not effectively mitigate the impact of pruning on the model's

---

*Corresponding authors: Ping Luo, pluo@cs.hku.hk; Wenqi Shao, shaowenqi@pjlab.org.cn

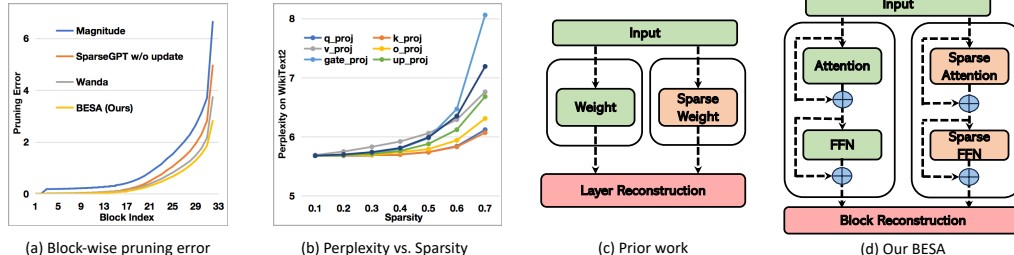

Figure 1: (a) shows that layer-wise pruning methods such as Wanda (Sun et al., 2023) produce a more significant error than our block-wise pruning technique BESA. (b) presents the curves of perplexity v.s. sparsity for different layers on WikiText2 (Merity, 2016). We see that layers do not contribute equally to the final performance. (c) shows that prior works prune all linear projections in the transformer block by layer reconstruction. (d) expresses that our proposed BESA compresses LLMs under a block-wise reconstruction pipeline.

output because the pruning error would accumulate layer by layer as demonstrated in Fig.1(a). Secondly, layer-wise pruning also requires handcrafting the sparsity for all layers, as the individual contributions of layers to the final model performance exhibit significant variation, as illustrated in Fig.1(b). Applying a uniform pruning rate to all layers, as seen in prior methods, poses the risk of removing important weights, given the unequal contributions of layers to the final performance.

To address these challenges, we propose the Blockwise Parameter-Efficient Sparsity Allocation (BESA) technique for compressing LLMs, which optimizes pruning rates across different layers as shown in Fig.1(d). Toward this goal, we first formulate the sparsity allocation problem to minimize block-wise reconstruction error with a learnable binary mask for each weight. BESA enjoys two advantages for LLM compression. Firstly, the sparsity that was previously considered a non-differentiable hyperparameter can be now equivalently represented by differentiable binary masks. Hence, layer-wise pruning sparsity can be optimized using a simple gradient descent algorithm. Secondly, unlike traditional approaches (Kang & Han, 2020) that learn sparsity for the entire model, BESA optimizes pruning rates sequentially within each transformer block. This enables efficient and differentiable pruning of LLMs ranging from 7B to 180B parameters on a single A100 GPU.

However, directly learning binary masks is challenging because it involves a huge solution space. To mitigate this issue, BESA encodes the fact that a more important weight would have a lower pruning probability in a parameter-efficient way (*e.g.* 2.10% extra parameters of a transformer block in LLaMA1-7B). The binary mask can be generated with element-wise pruning probabilities whose gradients are easily obtained through straight-through estimation (Bengio et al., 2013). Such a procedure remarkably reduces the solution space and alleviates the learning difficulty. We further develop a comprehensive LLM compression framework where weight pruning and quantization are jointly optimized in a differentiable manner. Extensive experiments show that BESA achieves state-of-the-art performance in pruning various LLMs such as LLaMA1 (Touvron et al., 2023a), and LLaMA2 (Touvron et al., 2023b).

Overall, this work has three contributions. (1) We propose a model pruning framework named BESA for compressing LLMs which searches for optimal pruning rates for each layer in a differentiable manner. To the best of our knowledge, BESA is the first differentiable pruning algorithm for LLMs. (2) Our BESA is parameter-efficient and easy to optimize, exhibiting high efficiency and effectiveness in pruning various LLMs such as LLaMA1, and LLaMA2. For example, BESA can prune 50% parameters of LLaMA2-70B (Penedo et al., 2023) within five hours on a single A100-80GB GPU with 0.16 perplexity improvement on WikiText2 (Merity, 2016) compared with SparseGPT (Frantar & Alistarh, 2023). (3) Extensive experiments on language modeling tasks such as WikiText2, PTB (Marcus et al., 1994), and C4 (Raffel et al., 2020) and various downstream tasks show that BESA establishes new state-of-the-art performance compared with prior arts. Finally, we demonstrate the practical speedup of the pruned model in a hardware simulator.

## 2 RELATED WORK

**Compression of Large Language Models.** Numerous technologies aim to mitigate the memory and computation demands of Large Language Models (LLMs). These techniques can be broadly

categorized into two primary types: quantization (Frantar et al., 2022; Lin et al., 2023; Shao et al., 2023) and pruning (Sun et al., 2023; Frantar & Alistarh, 2023; Ma et al., 2023). Quantization converts full-precision values to low-bit representations, while pruning selectively eliminates insignificant weights. These two compression strategies are distinct but can be synergistically combined to enhance the compression ratio (Frantar et al., 2022; Kim et al., 2023). In this paper, we focus on impelling the performance of LLM pruning.

**Pruning of Large Language Models.** Pruning methods for neural networks can be broadly classified into structured pruning (Ma et al., 2023; Huang et al., 2020) and unstructured pruning (Frantar et al., 2022; Sun et al., 2023; Zhang et al., 2023; 2022b). Conventional techniques such as those in (Huang et al., 2020; Zhang et al., 2023) are ill-suited for LLMs due to their reliance on extensive retraining, a challenge amplified within the context of LLMs. In contrast, LLM-specific pruning methods emphasize data and time efficiency. Regarding structured pruning, LLMpruner (Ma et al., 2023) delves into the structured pruning of LLMs and employs LoRA to recuperate the performance of pruned models. In the realm of unstructured pruning, SparseGPT (Frantar & Alistarh, 2023) introduces an efficient technique for estimating the Hessian matrix, thereby adapting the traditional OBS approach (Hassibi et al., 1993) to large-scale models. Furthermore, Wanda (Sun et al., 2023) adopts a straightforward strategy, eliminating weights based on the product of weight and activation values. Those methods prune LLMs using a layer-wise strategy and employ a consistent pruning rate throughout all layers, leading to rapid error accumulation. Contrarily, our approach emphasizes block-wise pruning, coupled with a differential allocation of layer-specific sparsity, which effectively minimizes performance degradation.

**Sparsity Allocation in Network Pruning.** Several previous methods (Chen et al., 2023; Kusupati et al., 2020; Evci et al., 2020)) have been proposed to achieve adaptive layer-wise sparsity. For instance, STR (Kusupati et al., 2020) ) and LATS (Chen et al., 2023)) introduce learning-based approaches to determine pruning thresholds for each layer, leading to a non-uniform sparsity distribution. However, directly adapting these techniques to LLMs presents challenges, primarily due to the extensive retraining needed on vast datasets. Our approach is tailored to efficiently address this issue.

## 3 METHOD

This section introduces our Blockwise Parameter-Efficient Sparsity Allocation (BESA) framework for compressing LLMs. As shown in Fig.2, our proposed BESA sequentially prunes the parameters of one transformer block before moving on to the next under the supervision of block-wise reconstruction error minimization. Such a pipeline reduces the GPU memory overhead remarkably. In addition, we develop a parameter-efficient sparsity learning algorithm to optimize sparsity for each layer in a block. We introduce the proposed BESA framework in the following. The overall algorithm is presented in Algorithm 1.

### 3.1 BLOCK-WISE PRUNING

BESA solves the optimization problem via block-wise pruning, making it possible to prune LLM with the parameter size of 7B - 180B on a single A100 GPU. To facilitate differentiable sparsity learning in the block-wise setting, our objective becomes minimizing the reconstruction error between the blockwise outputs of pruned and dense models as shown in Fig.2(a) and Fig.2(a).

For each transformer block, we drop the superscript '$l$' for simplicity of notation. In this way, block-wise pruning can be expressed as

$$\arg\min_{\mathcal{M}} \mathcal{L}^{\text{block}} = \underbrace{\|\mathsf{F}(\mathcal{W}, \mathbf{X}) - \mathsf{F}(\mathcal{W} \odot \mathcal{M}, \mathbf{X})\|_F^2}_{\mathcal{L}^{\text{recon}}} + \lambda \underbrace{\left(\frac{1}{T^b} \sum_{\mathbf{M} \in \mathcal{M}} \mathsf{k}(\mathbf{M}) - \hat{\alpha}\right)^2}_{\mathcal{L}^{\text{sparse}}} \quad (1)$$

where $\mathcal{W}$ and $\mathcal{M}$ are the set of all linear weights in self-attention and feed-forward modules and their corresponding learnable binary masks. $T^b$, $\mathbf{X}$, and $\mathsf{F}$ denote the total parameter size of the transformer block, input token, and the mapping function, respectively. $\mathsf{k}(\cdot)$ returns the number of zero entries and $\mathbf{M}$ is the binary mask for each linear weight whose zero entry indicates that the corresponding weight is pruned, $\|\cdot\|_F$ is the Frobenuous norm, and $\lambda$ is a hyperparameter.

In Eqn.(1), block-wise pruning is built with a reconstruction loss $L^{recon}$, which minimizes the pruning error, and a sparsity penalty loss $L^{sparse}$, which encourages the pruned model to satisfy the

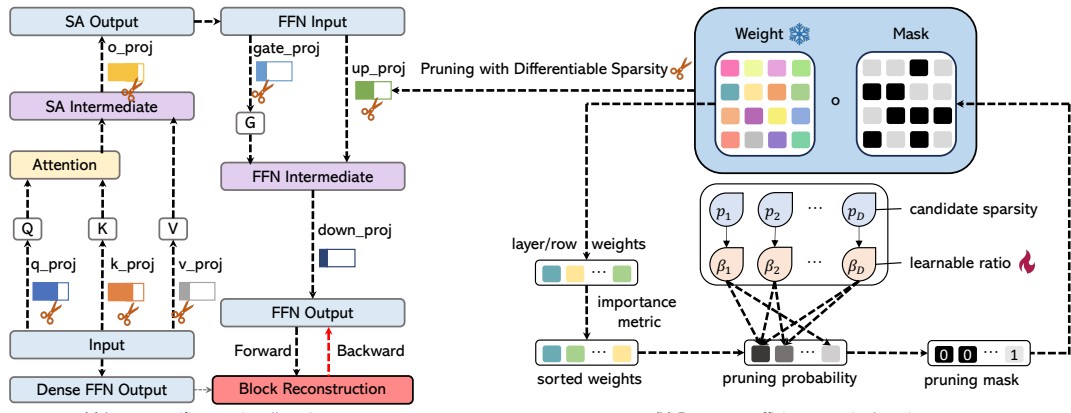

Figure 2: The pipeline of our BESA. (a) shows that BESA prunes weights in the self-attention and feed-forward networks by block reconstruction, which enables efficient and differentiable search for layer-specific pruning rates. (b) describes that weight pruning is achieved by differentiable binary masks which are obtained in a parameter-efficient way by taking weights' importance into modeling. Note that only a small number of ratios $\{\beta_d\}_{d=1}^D$ are learnable during pruning while the original weights in the LLM are frozen.

sparsity constraint. The sparsity penalty is instantiated with a $\ell_2$ loss, which we find works well to attain the target sparsity $\hat{\alpha}$ for each transformer block. The block-wise pruning in Eqn.(1) sequentially prunes the weights of one transformer block before moving on to the next. In this way, it is sufficient to guarantee the global sparsity of the whole LLM. Moreover, since each linear weight maintains a binary mask whose 0-1 values can be optimized through a gradient descent algorithm, our BESA can obtain the optimal sparsity for each linear weight.

Although BESA reduces the memory footprint overhead by block-wise pruning, it still requires learning binary masks $\mathcal{M}$ for all linear weights, which involves a huge solution space. Instead of directly learning binary masks with massive parameters, we develop a parameter-efficient algorithm to learn layer sparsity with marginally additional learnable parameters in Sec.3.2.

## 3.2 PARAMETER-EFFICIENT SPARSITY LEARNING

Our BESA employs a parameter-efficient sparsity learning technique to enable weight pruning with optimal pruning rate for LLMs. Motivated by the fact that pruning unimportant weights minimizes performance degradation, we propose to remove the top-$K$ least important weights for each layer. Note that $K$ can be different for layers which also implies that each layer has its own optimal sparsity $\alpha^*$ (i.e. $\alpha^* = K/N$ where $N$ denotes the parameter size of linear weights), considering that layers in a transformer block do not contribute equally to the final performance as shown in Fig.1(b).

To optimally select the top-$K$ least important weights for each layer, our main idea is to first sort weights with weight importance metric and then assign important (unimportant) weights with a mask 1 (mask 0) in a differentiable manner, as shown in Fig.2(b).

**Weight Sorting.** Various metrics have been proposed to measure the importance of weights of LLM. For example, SparseGPT (Frantar & Alistarh, 2023) estimates the importance of weight by the incurring error when the underlying weight is masked. Moreover, for each individual weight, Wanda (Sun et al., 2023) evaluates its importance by the product of weight magnitude and the corresponding input feature norm, which simplifies SparseGPT by avoiding calculating the Hessian inverse. Here, we directly adopt Wanda as the weight importance metric to sort weights.

Given layer weight $\mathbf{W} \in \mathbb{R}^{C_{in} \times C_{out}}$ and layer input $\mathbf{x} \in \mathbb{R}^{S \times C_{in}}$ where $C_{in}, C_{out}$ and $S$ are weight input dimension, weight output dimension, and input sequence length, respectively, we sort the weights of each row by

$$\delta_{i,j} = |W_{i,j}| \cdot ||\mathbf{x}_{:,j}||_2, \ W_{i\hat{j}} = \text{Sort}(W_{i,j}|\delta_{i,j}) \tag{2}$$

where $W_{i,j}$ is $i$-th row and $j$-th column entry of $\mathbf{W}$ and $\mathbf{x}_{:,j}$ of the $j$-th column vector of $\mathbf{x}$. The weight importance $\delta_{i,j}$ takes both weight and activation magnitude into consideration. It works well

in our BESA to find the least top-$K$ weights. With $\delta_{i,j}$, we can obtain the sorted weight sequence $w_{i,\hat{j}}$ in ascending order by the $\text{Sort}(\cdot)$ function. We also experiment with other metrics of weight importance in Appendix Sec.A. Note that we only need to sort weights of each row in each layer once by Eqn.(2) as shown in Algorithm 1, considering that the weight's importance is invariant to the pruning procedure.

**Mask Generation.** We now turn to generate differentiable masks in a parameter-efficient way. Towards this goal, we parameterize the sparsity with the learnable combination of a set of candidate pruning rates $\{p_d\}_{d=1}^{D}$ where $p_d \leq p_{d+1}$ and $D$ denotes the number of candidate pruning rates. In this way, we formulate the layer sparsity as

$$\alpha = \sum_{d=1}^{D} \beta_d p_d, \tag{3}$$

where $\boldsymbol{\beta} = [\beta_1, \cdots, \beta_D]^T \in \Delta^{D-1}$ are learnable coefficients lying in a simplex and $\beta_d$ is the probability of layer sparsity being $p_d$. Note that the top-$(C_{out} \cdot p_d)$ least important will be pruned if the layer sparsity is $p_d$. Given candidate pruning rates $\{p_d\}_{d=1}^{D}$, we can derive the element-wise weight pruning probability as

$$P(W_{i,\hat{j}}) = \sum_{d=k+1}^{D} \beta_d \quad \text{if } C_{out} \cdot p_k \leq \hat{j} < C_{out} \cdot p_{k+1} \tag{4}$$

where $P(W_{i,\hat{j}})$ indicates the probability that weight $W_{i,\hat{j}}$ is pruned. We set the boundary condition as $p_0 = 0$ and $\beta_D = 0$ which ensures that the most important weights are always retained. From Eqn.(4), we have $P(W_{i,\hat{j}}) \geq P(W_{i,\hat{j}+1})$. Hence, our modeling of element-wise weight pruning probability explicitly encodes the fact that the less important weights have higher pruning probability, which would reduce the optimization difficulty in finding unimportant weights. Given the pruning probability of each weight, the weight mask $\mathbf{M}$ in Eqn.(1) can be generated by

$$M_{i,j} = \begin{cases} 0, \text{if } P(M_{i,j}) \geq \alpha, \\ 1, \text{esle,} \end{cases} \tag{5}$$

where $M_{i,j} = 1$ indicates that the weight $W_{i,\hat{j}}$ is preserved, and vice versa.

**Differentiability of Sparsity** $\alpha$**.** Our modeling of binary mask $\mathbf{M}$ make the loss function $\mathcal{L}^{block}$ differentiable with respect to sparsity $\alpha$. Note that the gradient of mask $M_{i,j}$ with respect to pruning probability $P(W_{i,j})$ can be estimated using Straight-Through-Estimator (STE). Hence, the gradient w.r.t. $\alpha$ can be calculated by

$$\frac{\partial \mathcal{L}^{block}}{\partial \alpha} = p_d \sum_{d=1}^{D} \frac{\partial \mathcal{L}^{block}}{\partial \beta_d}, \frac{\partial \mathcal{L}^{block}}{\partial \beta_d} = \sum_{j=1}^{C_{out}} \frac{\partial \mathcal{L}^{block}}{\partial M_{i,j}} \frac{\partial P(M_{i,j})}{\partial \beta_d} \tag{6}$$

With Eqn.(6), the sparsity (pruning rate) can be optimized through a simple gradient descent algorithm for different layers.

**Parameter Efficiency.** The learnable parameters of the sparsity learning in Eqn.(1 - 5) come from the combination coefficients $\{\beta_d\}_{d=1}^{D}$. By default, we learn sparsity for each row, which results in additional $D \cdot C_{in}$ parameters for each layer. However, learning sparsity on a row basis would cause an unstructured pruning mask, which slows down the learning process because unstructured masks cannot be implemented in a parallel way. To address this, we have designed a customized CUDA operator to accelerate the row-wise probability pruning mask generation in this setting. We also implement a lightweight version with fewer learnable parameters by sharing the same set of combination coefficients $\{\beta_d\}_{d=1}^{D}$, which adds only $D$ parameters for each layer. In experiments, we set $D = 100$. Take LLaMA-65B as an example, our BESA introduces $2.10\%$ and $0.0003\%$ extra parameters in each block for row-wise and layer-wise settings, respectively.

### 3.3 JOINT OPTIMIZATION WITH QUANTIZATION

Pruning can save memory by only storing unpruned weights and binary masks while quantization reduces memory by saving weights in the low-bit format. Thanks to the inclusion of block-wise reconstruction, our BESA pruning algorithm can be jointly optimized with the weight-only quantization technique. Following OmniQuant (Shao et al., 2023), we consider the Min-Max quantization scheme with learnable clipping strengths. To jointly optimize quantization and pruning parameters,

---

**Algorithm 1** Overall algorithm of BESA.

---

**Input**: calibration dataset $\mathcal{X}$, pre-trained LLM model $\{\mathcal{W}^l\}_{l=1}^L$, and target sparsity $\hat{\alpha}$.
**Output**: pruned model.

1: Initialize $\mathbf{X}_p = \mathcal{X}$,              ▷ init inputs of pruned model.
2: **for** $l$ in $\{1, 2, \cdots, L\}$ **do**:              ▷ block-wise pruning
3:     calculate the full-precision output $\mathsf{F}(\mathbf{X}_{fp}, \mathcal{W}^l)$,
4:     sort weights for all $\mathbf{W} \in \mathcal{W}^l$ by Eqn.(2),      ▷ only sort weights once for each block
5:     **while** optimal sparsity $\alpha_l^*$ not converge **do**:
6:         generate element-wise pruning mask $M_{i,j}^l$ with learnable ratios $\{\beta_d\}_{d=1}^D$ by Eqn.(3 - 5),
7:         calculate pruned output $\mathsf{F}(\mathbf{X}_p, \mathcal{W}^l \odot \mathcal{M}^l)$,
8:         calculate block loss $\mathcal{L}^{\text{block}}$ by Eqn.(1),
9:         update learnable ratios $\{\beta_d\}_{d=1}^D$ by back-propagation,
10:     **end while**
11:     forward propagation $\mathbf{X}_p = \mathsf{F}(\mathbf{X}_p, \mathcal{W}^l \odot \mathcal{M}^l)$,
12:     store the weight mask $\mathcal{M}^l$ ,
13: **end for**
14: **return** pruned model $\{\mathcal{W}^l \odot \mathcal{M}^l\}_{l=1}^L$.

---

we first quantize the model weights and then prune the quantized weights with our BESA. This can be implemented by substituting $\mathcal{W}$ with its quantized version $\mathsf{Q}(\mathcal{W})$ which is expressed as

$$\mathsf{Q}(\mathbf{W}) = \text{clamp}(\lfloor \frac{\mathbf{W}}{h} \rceil + z), \text{ with } h = \frac{\gamma_1 \max(\mathbf{W}) - \gamma_0 \min(\mathbf{W})}{2^N - 1}, z = -\lfloor \frac{\gamma_0 \min(\mathbf{W})}{h} \rceil \quad (7)$$

for all $\mathbf{W} \in \mathcal{W}$. In Eqn.(7), $\mathsf{Q}(\mathbf{W})$ and $\mathbf{W}$ denote the quantized and full-precision weights, respectively. $h$ is the normalization factor for weights and $z$ is the zero-point value. The clamp operation constrains the value within the range of $N$-bit integer, specifically $[0, 2^N - 1]$ where $N$ is the target bit number. $\lfloor \cdot \rceil$ indicates round operation. $\max(\mathbf{W})$ and $\min(\mathbf{W})$ are the maximum and minimum in $\mathbf{W}$, respectively. $\gamma_0 \in [0, 1]$ and $\gamma_1 \in [0, 1]$ are learnable clipping strengths for the lower and the upper bound of weights, respectively. When performing pruning and quantization simultaneously, we optimize the combination coefficients $\{\beta_d\}_{d=1}^D$ for generating pruning masks and quantization clipping thresholds $\{\gamma_0, \gamma_1\}$.

## 4 EXPERIMENTALS

In this section, we present a comprehensive series of experiments designed to evaluate the effectiveness of our proposed methods. We begin by providing a detailed overview of our experiment settings, encompassing the configuration of our experiments, the specific Large Language Model (LLM) model under evaluation, the benchmark dataset utilized, and the baseline method employed for comparison. Subsequently, we assess both the perplexity and the zero-shot capability of the pruned LLM models. Finally, we concurrently perform pruning and quantization, and we include a series of ablation studies, which can be found in Appendix Sec.A. Additionally, we explore the real-world acceleration performance of our proposed method using a customized accelerator known as ViTCoD (You et al., 2023).

### 4.1 EXPERIMENT SETTINGS

**Setup.** All pruning experiments were executed on a single NVIDIA A100 GPU equipped with 80GB of memory. Our proposed method, along with the baseline methods, was implemented using the PyTorch framework. The calibration set used consisted of 128 sequences, each comprising 2048 tokens, sampled from the first shard of the C4 training dataset, mirroring the approach adopted in the baseline methods. LLM models and datasets were sourced from the Huggingface Transformers library (Wolf et al., 2020). Zero-shot experiments were conducted with the assistance of the Language Model Evaluation Harness (LM-Eval) library (Gao et al., 2021). In this configuration, our proposed method achieved full sparsity in the LLaMA-65B model within a remarkable time frame of 4.5 hours.

| Datasets | Methods | 1-7B | 1-13B | 1-30B | 1-65B | 2-7B | 2-13B | 2-70B |
|---|---|---|---|---|---|---|---|---|
| | Dense | 5.68 | 5.09 | 4.10 | 3.53 | 5.47 | 4.88 | 3.31 |
| Wikitext2 | SparseGPT | 7.22 | 6.21 | 5.33 | 4.60 | 6.99 | 6.02 | 4.25 |
| | Wanda | 7.26 | 6.15 | 5.25 | 4.60 | 6.92 | 5.97 | 4.22 |
| | BESA | **6.86** | **5.92** | **5.00** | **4.33** | **6.60** | **5.75** | **4.09** |
| | Dense | 7.34 | 6.70 | 6.13 | 5.81 | 7.26 | 6.73 | 5.71 |
| C4 | SparseGPT | 9.31 | 8.12 | 7.33 | 6.66 | 9.23 | 8.22 | 6.45 |
| | Wanda | 9.34 | 8.14 | 7.29 | 6.71 | 9.24 | 8.30 | 6.50 |
| | BESA | **8.96** | **7.90** | **7.09** | **6.49** | **8.88** | **7.96** | **6.38** |
| | Dense | 41.25 | 28.10 | 23.51 | 25.07 | 32.91 | 48.82 | 20.76 |
| PTB | SparseGPT | 79.25 | 37.24 | 26.33 | 27.93 | 108.71 | 70.87 | 22.67 |
| | Wanda | 80.30 | 36.42 | 26.63 | 25.75 | 48.15 | 69.65 | 23.20 |
| | BESA | **66.96** | **36.07** | **25.41** | **24.76** | **44.09** | **58.58** | **22.87** |

Table 1: Perplexity results for LLaMA models with unstructured 50% sparsity. In the table, 1-7/13/30/65B denotes LLaMA-7/13/30/65B, and 2-7/13/70B represents LLaMA2-7/13/70B models. The **best** performing result is indicated in **bold**, while the second best result is shown as underlined.

**Models.** Our primary focus for evaluation centers on the LLaMA (Touvron et al., 2023a) family of models, renowned as one of the most prominent series of Large Language Models (LLMs). Specifically, we assessed our methods across various model sizes, including LLaMA-7B/13B/30B/65B, and LLaMA2-7B/13B/70B. Notably, our methodology exhibits consistent improvements and is not bound by the size of the LLaMA model.

**Benchmarks.** Our initial assessment centers on evaluating the perplexity of pruned LLM models, a widely recognized metric renowned for its reliability and resilience in appraising LLM performance. In alignment with prior studies (Frantar & Alistarh, 2023; Sun et al., 2023), we primarily measure model perplexity using the WikiText2 (Merity, 2016), C4 (Raffel et al., 2020), and PTB (Marcus et al., 1994) datasets. In addition to assessing perplexity, we undertake an exhaustive examination of the zero-shot capabilities of pruned models across six standard common-sense benchmark datasets. These benchmarks encompass PIQA (Bisk et al., 2020), BoolQ (Clark et al., 2019), HellaSwag (Zellers et al., 2019), WinoGrande (Sakaguchi et al., 2021), as well as the ARC Easy and Challenge (Clark et al., 2018) tasks.

**Baselines.** We evaluate the following established methods as baselines: (i) SparseGPT, which divides the task of pruning LLM models into a sparse regression problem for a set of transformer blocks, subsequently solving these problems with an approximate sparse regression solver. It is worth noting that SparseGPT updates the values of unpruned weights. (ii) Wanda, a method that leverages the product of weight magnitude and L2 normalization of input activations to determine the importance of LLM model weights, followed by pruning weights with lower importance.

## 4.2 PERPLEXITY EXPERIMENTS

In this experimental evaluation, we conducted a comprehensive assessment of the entire LLaMA model family. We pruned all linear layers, excluding embeddings and the model head, achieving a 50% unstructured sparsity level. The perplexity scores for the pruned models on the Wikitext2, C4, and PTB datasets are presented in Table 1. The results displayed in Table 1 demonstrate a consistent improvement in BESA when compared to existing methods. To further explore the impact of different sparsity levels, we conducted experiments with varying sparsities, and the results, measured in terms of Wikitext2 perplexity, are visualized in Fig.3.

## 4.3 ZERO-SHOT EXPERIMENTS

In addition to utilizing perplexity as a reliable and robust metric for assessing LLM performance, we have expanded our evaluation to encompass a range of downstream prompt-based zero-shot tasks. We provide a concise summary of detailed performance metrics in Table 2. When considering

| Models | Methods | PIQA | BoolQ | HellaSwag | Winogrande | ARC-e | ARC-c | Average |
|---|---|---|---|---|---|---|---|---|
|  | Dense | 78.67 | 75.08 | 56.94 | 70.01 | 75.25 | 41.89 | 66.31 |
| 1-7B | SparseGPT | 76.39 | **72.97** | 51.41 | **69.38** | **71.30** | **37.29** | 63.12 |
|  | Wanda | 75.41 | 71.04 | 51.95 | 66.14 | 69.36 | 36.95 | 61.81 |
|  | BESA | **76.66** | 72.17 | **54.31** | 67.64 | 70.79 | 37.20 | **63.13** |
|  | Dense | 79.16 | 77.89 | 59.93 | 72.69 | 77.36 | 46.42 | 68.91 |
| 1-13B | SparseGPT | **78.35** | 76.85 | 54.88 | 71.35 | 72.47 | 41.98 | 65.98 |
|  | Wanda | 77.42 | 76.27 | 55.77 | **72.30** | 73.32 | 43.86 | 66.49 |
|  | BESA | 77.97 | **76.91** | **57.61** | 72.06 | **73.86** | **46.16** | **67.43** |
|  | Dense | 81.07 | 82.72 | 63.34 | 75.93 | 80.43 | 52.90 | 72.73 |
| 1-30B | SparseGPT | 79.65 | 82.87 | 59.21 | 73.64 | 78.91 | 48.89 | 70.53 |
|  | Wanda | 79.33 | 81.87 | 60.96 | 73.88 | 79.38 | **50.09** | 70.92 |
|  | BESA | **79.82** | **83.12** | **62.39** | **75.06** | **79.67** | 49.57 | **71.61** |
|  | Dense | 81.23 | 84.83 | 64.55 | 77.43 | 81.31 | 52.90 | 73.71 |
| 1-65B | SparseGPT | 80.52 | 85.08 | 62.21 | **77.82** | 79.88 | 50.26 | 72.63 |
|  | Wanda | 80.63 | 85.47 | 62.77 | 77.43 | 80.26 | 50.34 | 72.82 |
|  | BESA | **80.74** | **85.54** | **64.35** | 77.27 | **81.10** | **53.38** | **73.73** |
|  | Dense | 78.07 | 77.71 | 57.14 | 68.90 | 76.35 | 43.60 | 66.96 |
| 2-7B | SparseGPT | 76.17 | **76.02** | 52.81 | **68.67** | 71.63 | 36.95 | 63.71 |
|  | Wanda | 76.55 | 75.29 | 52.65 | 67.17 | 72.18 | 38.99 | 63.81 |
|  | BESA | **76.66** | 74.83 | **54.60** | 68.59 | **73.86** | **40.96** | **64.92** |
|  | Dense | 79.05 | 80.55 | 60.06 | 72.14 | 79.42 | 48.46 | 69.95 |
| 2-13B | SparseGPT | 77.69 | 81.41 | 55.93 | **71.59** | 74.66 | 42.06 | 67.22 |
|  | Wanda | 78.62 | 81.04 | 56.97 | 71.51 | 76.26 | **43.26** | 67.94 |
|  | BESA | **79.11** | **81.68** | **59.19** | 70.80 | **76.64** | **43.26** | **68.45** |
|  | Dense | 82.21 | 83.79 | 64.77 | 77.90 | 82.70 | 54.44 | 74.30 |
| 2-70B | SparseGPT | 81.56 | 85.05 | 62.23 | **78.30** | **81.65** | 53.33 | 73.69 |
|  | Wanda | 81.01 | 83.24 | 62.66 | 77.19 | 81.14 | 52.05 | 72.88 |
|  | BESA | **81.72** | **85.38** | **63.81** | 77.27 | 81.52 | **53.41** | **73.85** |

Table 2: LLaMA accuracies for zero-shot tasks with unstructured 50% sparsity. In the table, 1-7/13/30/65B denotes LLaMA-7/13/30/65B, and 2-7/13/70B represents LLaMA2-7/13/70B models. The **best** performing result is indicated in **bold**, while the second best result is shown as underlined.

the average accuracy across the six tasks we examined, BESA consistently demonstrates superior performance compared to existing methods. Notably, the disparity in average accuracy between our pruned model and the original dense model diminishes as the model size increases. While it is important to acknowledge that the evaluation results for these prompt-based zero-shot tasks exhibit more variability compared to perplexity, BESA even achieves higher average accuracy than the original dense model in LLaMA-65B.

## 4.4 JOINT COMPRESSION

We explore the synergy of combining both pruning and quantization techniques. Introducing sparsity into quantized models enhances their potential for achieving significant gains in terms of speed and memory efficiency, thereby facilitating the deployment of LLMs on edge devices. As detailed in Sec.3.3, we have harnessed the cutting-edge OmniQuant method (Shao et al., 2023) to implement 4-bit weight-only quantization in conjunction with our pruning algorithm, employing a block-wise approach. The performance of the jointly compressed models in LLaMA-7/13/30B and LLaMA2-7/13B is presented in Table 3. For the sake of comparison, we have also applied post-pruning to the quantized model using the Wanda method. As demonstrated in Table 3, under the joint compression framework, BESA consistently outperforms Wanda across various models and datasets.

| Models | Wikitext2 | | | C4 | | | PTB | | |
|--------|-------|-------|-------------|-------|-------|-------------|-------|-------|-------------|
| | Dense | Joint | Joint-Wanda | Dense | Joint | Joint-Wanda | Dense | Joint | Joint-Wanda |
| 1-7B   | 5.68 | 7.00 | 7.44 | 7.34 | 9.16 | 9.64 | 41.25 | 73.14 | 92.79 |
| 1-13B  | 5.09 | 6.01 | 6.27 | 6.70 | 8.02 | 8.30 | 28.10 | 35.43 | 36.30 |
| 1-30B  | 4.10 | 5.08 | 5.34 | 6.13 | 7.20 | 7.44 | 23.51 | 25.63 | 27.11 |
| 2-7B   | 5.47 | 6.77 | 7.12 | 7.26 | 9.10 | 9.50 | 32.91 | 49.91 | 53.26 |
| 2-13B  | 4.88 | 5.85 | 6.10 | 6.73 | 8.07 | 8.44 | 48.82 | 61.17 | 71.10 |

Table 3: Perplexity (ppl) Evaluation of LLaMA Family Models with Joint Compression (lower ppl indicates superior performance). In this table, Dense refers to the original dense model, Joint corresponds to the outcomes achieved through concurrent BESA-based pruning and quantization, and Joint-Wanda signifies the results obtained by pruning the quantized model with Wanda.

| Layer name | q_proj | k_proj | v_proj | o_proj | gate_proj | up_proj | down_proj |
|------------|--------|--------|--------|--------|-----------|---------|-----------|
| Dense Runtime | 4096 | 4096 | 4096 | 4096 | 10128 | 10128 | 10128 |
| Average Runtime (SparseGPT) | 2952.22 | 2932.0 | 3041.31 | 2950.13 | 7941.88 | 7865.81 | 7441.44 |
| Average Runtime (Wanda) | 2887.84 | 2871.34 | 3000.91 | 2461.59 | 7701.41 | 7670.84 | 7388.97 |
| Average Runtime (BESA) | 2232.31 | 2230.50 | 2720.59 | 2698.53 | 5207.53 | 5125.0 | 6850.03 |
| BESA Sparsity | 53.87% | 54.54% | 48.96% | 47.15% | 50.20% | 50.36% | 46.52% |
| BESA Speedup | 1.83× | 1.84× | 1.51× | 1.52× | 1.94× | 1.98× | 1.48× |

Table 4: Runtime (cycles) and speedup across various layer shapes in LLaMA-7B. The term "cycles" denotes the number of instruction cycles necessary for the ViTCoD accelerator to perform the associated computational workloads.

## 4.5 Speedup in Simulation

Prior unstructured pruning techniques (Frantar & Alistarh, 2023; Sun et al., 2023) exploit a fine-grained structured sparsity scheme (*e.g.* $n : m$ sparsity), to achieve acceleration on real computing devices. The $n : m$ technique can be effectively implemented on NVIDIA Ampere GPUs using the cuSPARSELt library to achieve practical speed improvements. Our BESA seeks the optimal pruning rate for each layer. which yet poses challenges in achieving the structured $n : m$ sparsity pattern. To comprehensively investigate the speedup potential of pruned Large Language Models (LLMs), we have utilized specialized neural network accelerators other than NVIDIA GPUs.

Specifically, we employ the simulator of ViTCoD accelerator (You et al., 2023), to assess the realistic speedup capabilities of our proposed method. The ViTCoD accelerator incorporates a denser and sparser engine, designed to concurrently process computation workloads with varying levels of sparsity. This simultaneous processing enhances the utilization of Processing Elements (PEs) within the accelerator. In this work, we extend the capabilities of ViTCoD to handle the computation of all sparse matrix multiplications within a pruned transformer block. We provide more details about the configurations of ViTCoD in Appendix Sec.B.

Given that sparsity significantly influences the runtime of computation, and considering that our BESA prunes the model with imbalanced layer sparsity within each transformer block, we calculate the average simulated runtime across all transformer blocks within LLaMA-7B. Detailed speedup values for each pruned layer within the transformer block, along with their corresponding average sparsity, are provided in Table 4, accompanied with the simulated runtime of the model pruned by SparseGPT and Wanda for comparison.

## 5 Conclusion

In this work, we propose blockwise parameter-efficient sparsity allocation (BESA), which is a comprehensive framework to jointly prune and quantize large language models (LLM). We find that layer- wise pruning error minimization adopted by previous arts does not effectively mitigate the impact of pruning on the model's output because the pruning error would accumulate layer by layer. By contrast, our PESA operates under a blockwise pruning framework. By minimizing block-wise error and optimizing sparsity rates across layers, BESA is able to prune various LLMs such as LLaMA1, and LLaMA2. Our experiments show that BESA achieves state-of-the-art performance, with a moderate performance drop compared with the unpruned one.

ACKNOWLEDGMENTS

This paper is partially supported by the National Key R&D Program of China No.2022ZD0161000 and the General Research Fund of Hong Kong No.17200622.

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

# APPENDIX

## A    ABLATION STUDIES

In this section, we conduct ablation studies to comprehensively investigate the performance scalability of our method. We delve into how different pruning configurations influence the performance of our pruned model. To expedite experimentation and obtain results more efficiently, we focus on the LLaMA-7B model with an unstructured sparsity level of $50\%$ in the following trials.

**Calibration Size.** Our initial investigation centers on assessing how the performance of our pruning methods varies with different sizes of calibration data. The results, as measured by Wikitext2 perplexity, are presented graphically in Fig.4. Notably, BESA demonstrates the ability to achieve satisfactory results even with a limited number of calibration samples. With fewer than 64 calibration samples, increasing the calibration dataset size leads to a significant improvement; however, this improvement tapers off rapidly after reaching 64 calibration samples. For example, increasing the number of calibration samples from 128 to 256 only results in a marginal decrease of 0.02 in Wikitext2 perplexity.

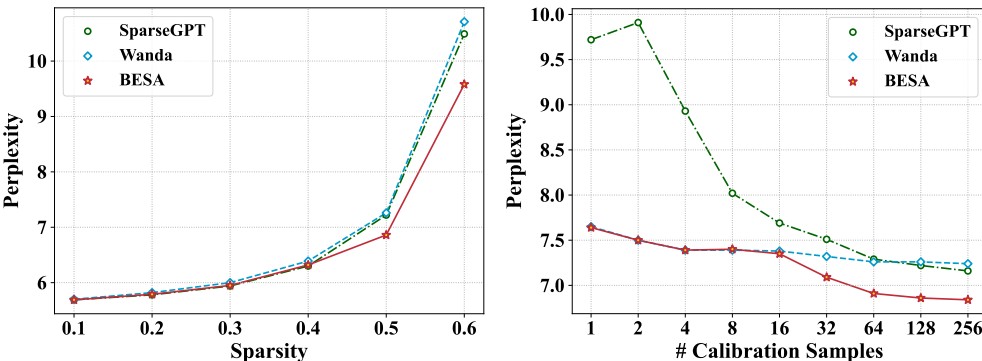

Figure 3: Model sparsity ablation             Figure 4: Calibration size ablation

**Epochs and Sparsity Steps.** Next, we investigate the influence of training epochs and sparsity steps on our pruning methodology. Detailed performance results are presented in Table 5. Given that the calibration data is drawn from the C4 dataset, we observe that the C4 perplexity of the pruned model decreases with an increasing number of training epochs. However, it is noteworthy that this trend is not consistently mirrored in the Wikitext2 and PTB datasets, suggesting that a larger number of training epochs may not necessarily yield a superior pruned model. Consequently, we adopt 1 epoch of training as our default setting, as it consistently produces pruned models with satisfactory perplexity across various datasets.

Then, we explore the significance of sparsity steps, which determine the sparsity candidates used in our method. For example, a sparsity step of 0.01 implies sparsity candidates ranging from 1.0 to 0.0 with a step size of 0.01. In Table 5, we compare the performance of three different sparsity steps: 0.1, 0.01, and 0.001. Notably, the C4 perplexity of the pruned model ceases to improve beyond a sparsity step of 0.01, prompting us to select it as our default sparsity step. Despite the seemingly better Wikitext2 perplexity associated with a 0.1 sparsity step, we opt for 0.01 for two primary reasons: i) Training with a 0.1 sparsity step requires more manual tuning to achieve model convergence at the target sparsity level. ii) Given that the calibration set is drawn from the C4 dataset and training with a 0.1 sparsity step results in higher C4 perplexity, it performs less favorably than other options in block-wise reconstruction.

Table 5: Ablation across epochs (left), sparsity steps (middle), and importance metrics (right).

| Epochs | 1 | 3 | 10 | 30 |
|---|---|---|---|---|
| Wikitext2 | 6.86 | 6.85 | 6.84 | 6.86 |
| C4 | 8.96 | 8.95 | 8.95 | 8.94 |
| PTB | 66.96 | 67.37 | 66.83 | 67.09 |

| Sparsity Step | 0.1 | 0.01 | 0.001 |
|---|---|---|---|
| Wikitext2 | 6.84 | 6.86 | 6.86 |
| C4 | 8.98 | 8.96 | 8.96 |
| PTB | 69.29 | 66.96 | 66.52 |

| Metric | Weight | Wanda | SparseGPT |
|---|---|---|---|
| Wikitext2 | 7.43 | 6.86 | 8.73 |
| C4 | 9.81 | 8.96 | 11.32 |
| PTB | 83.60 | 66.96 | 140.60 |

**Learning Granularity.** Another pivotal facet of our method concerns the selection of learning granularity. As depicted in Fig.1, the contribution of weights to the final performance exhibits significant variations across different layers. Previous methodologies (Frantar & Alistarh, 2023; Sun et al., 2023) have consistently applied a uniform pruning rate to all layers, thus introducing the peril of eliminating critical weights and detrimentally impacting overall performance. Consequently, we conducted an ablation study to meticulously assess the influence of learning granularity, as succinctly summarized in Table 6. We rigorously explored three distinct choices for learning granularity: Attn-MLP, block, and two blocks. Within the Attn-MLP setting, we permitted the sparsity of the Attention module and MLP module to converge to the target sparsity without imposing constraints on the individual layers within these modules. Analogously, the choices of block and two blocks followed similar principles.

Drawing upon the insights derived from Table 6, it becomes evident that larger learning granularity holds greater potential for preserving essential weights within the model, thereby leading to demonstrable performance enhancements. Furthermore, we delved into an in-depth investigation of the reconstruction errors associated with each block, corresponding to different learning granularities. These results are meticulously presented in Fig.5. Considering the combined improvements observed in perplexity, the reduction in reconstruction error, and the memory requirements associated with various learning granularities, we judiciously select the block as our default learning granularity.

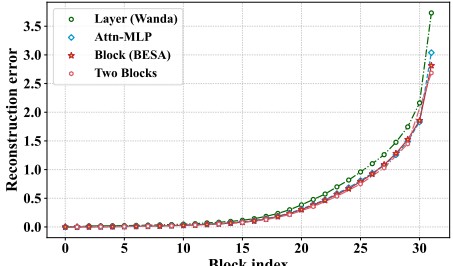

Table 6: Learning granularity ablation.

| Granularity | Wikitext2 | C4 | PTB |
|---|---|---|---|
| Layer (Wanda) | 7.26 | 9.30 | 80.30 |
| Attn-MLP | 6.97 | 9.14 | 70.70 |
| Block (BESA) | 6.86 | 8.96 | 66.96 |
| Two Blocks | 6.80 | 8.85 | 62.55 |

Figure 5: Reconstruction error for learning granularities.

**Importance Metric.** Finally, we ascertain the most suitable importance metric for our proposed method. Given that our method learns pruning probabilities based on sorted weights, we explore the impact of various importance metrics, with the results presented in Table 5. Notably, our method exhibits sensitivity to the choice of importance metrics. Consequently, we opt for Wanda's importance metric, which consistently delivers superior performance.

## B THE COMPUTATION MECHANISM OF VITCOD ACCELERATOR

As previously mentioned in Sec.4.5, our evaluation incorporates the ViTCoD accelerator (You et al., 2023) to assess the practical speedup capabilities of our proposed method. When considering the pruned weight matrix as a sparse matrix and the input activations as a dense matrix, the matrix multiplication performed in our pruned model can be characterized as a form of sparse-dense matrix multiplication (SpMM), as illustrated in Fig.6. ViTCoD addresses the challenges of SpMM computation, as depicted in Fig.7. This approach employs an output-stationary dataflow to reduce on-chip buffer requirements and minimize the need for frequent input matrix loading. Specifically, it spatially tiles both the sparse and dense matrices along the dimension shown in Fig.6 and accumulates partial sums along the dimension illustrated in the same figure. During computation, ViTCoD initially transfers the tiled blocks of input matrices to memory buffers and subsequently assigns computation tasks to either the Denser or Sparser Engine based on the sparsity of the tiled matrix columns. The partial sums computed in the Denser Engine are then transferred to the Sparser Engine and accumulated within the Sparser Engine's accumulator. This tiling and computation mapping strategy efficiently reuses input matrices and requires only a small on-chip buffer for storing calculated outputs. Furthermore, distributing computation workloads to the Denser and Sparser Engines enhances the utilization of Processing Elements (PEs).

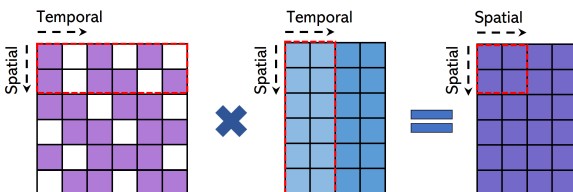

Figure 6: Illustration of Sparse-Dense Matrix Multiplication (SpMM). The leftmost matrix represents the sparse matrix, with zero values denoted by white blocks. The red dashed box highlights ViTCoD's tiling strategy for SpMM computation, while the spatial/temporal arrows illustrate ViT-CoD's computation mapping strategy.

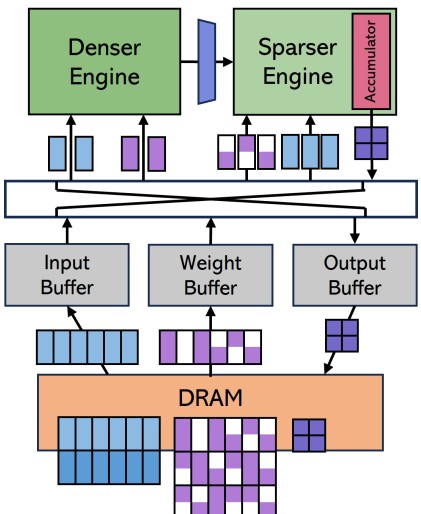

Figure 7: ViTCoD's mechanism for addressing SpMM computation.

