# OpenReview forum: "BESA: Pruning Large Language Models with Blockwise Parameter-Efficient Sparsity Allocation"
_ICLR.cc/2024/Conference — ICLR 2024 poster_

### Official Review · Reviewer_MvCd · 2023-10-18

**Soundness:** 3 good
**Presentation:** 2 fair
**Contribution:** 3 good
**Rating:** 8
**Confidence:** 2

**Summary:**

The authors proposed a novel method for blockwise pruning of LLMs. It was evaluated with generation tasks, as well as for zero-shot downstream tasks, and outperformed SparseGPT and Wanda baselines.

**Strengths:**

- The proposed method outperformed recent baselines.
- The motivation for this research is clear, and the proposed method is helpful for practitioners.

**Weaknesses:**

Authors claimed BESA has different advantages compared to other baselines. E.g., the fact that BESA differentiably optimizes masks, unlike SparseGPT. However, it is not clear whether other methods could or could not use such specific techniques and what makes BESA better than them. The current ablation study does not answer these questions.

**Questions:**

Please refer to the weaknesses Section.

---

> ### Author Response · Authors · 2023-11-20
> **Response to Reviewer MvCd**
>
> Thank you for your thoughtful feedback. We appreciate your engagement with our rebuttal and the points you've raised. We have carefully considered each of your concerns and addressed them as follows:
>
> **Q1:** "Authors claimed BESA has different advantages ... does not answer these questions."
>
> **A1:** Thank you for your insightful suggestion. In response, we have conducted experiments using three additional pruning techniques commonly employed in computer vision and natural language processing. These methods, namely iterative pruning (proposed in the Lottery Ticket Hypothesis [A]), threshold pruning [B], and direct mask learning, have been implemented in a blockwise manner for a fair comparison. For iterative pruning, we employ a step-wise approach where we iteratively prune the LLM block with a step of 10% according to the metric Wanda or Weight magnitude. During each iteration, we update the weights with the reconstruction loss. The results are reported in the table under "Iterative Prune". Then for threshold pruning, we learn a threshold value for each layer within the block while maintaining the block sparsity at 50%. The results are reported in the table under "Threshold Prune". For direct mask learning, we directly learn a sparse mask for the LLM block. The results are reported in the table under "Direct Mask Learning". We opt for iterative pruning as it stands out as one of the most representative methods for pruning, incorporating weight updating. Additionally, we select threshold pruning for its cutting-edge capabilities, showcasing state-of-the-art performance in computer vision models. Lastly, we choose direct mask learning as it represents the most straightforward and directly differentiable pruning method. Notably, the original “Iterative prune” and “Threshold prune” methods use the weight value as a measure of importance. In contrast, we provide the versions that use the weight or Wanda importance metric.
>
> The perplexity results on WikiText2, C4, and PTB datasets are presented below:
>
> |  | BESA | Iterative prune (Wanda) | Iterative prune (Weight) | Threshold prune (Wanda) | Threshold prune (Weight) | Direct mask learning |
> | --- | --- | --- | --- | --- | --- | --- |
> | WikiText2 ppl | 6.86 | 6.88 | 123.48 | 2984.61 | 20351.98 | 1457.53 |
> | C4 ppl | 8.96 | 9.08 | 102.11 | 2191.59 | 21203.34 | 371.47 |
> | PTB ppl | 66.96 | 57.71 | 3929.01 | 4600.38 | 32474.87 | 3927.83 |
>
> These results underscore the unique advantages of our method, particularly its ability to achieve substantial reductions in perplexity. We believe that this thorough exploration of differentiable methods provides a comprehensive understanding of the strengths and capabilities of BESA compared to other baselines.
>
> Additionally, we have re-implemented our algorithm in the style of structured pruning and compared it with LLM-Pruner [C], the latest structured pruning algorithm we found. The comparison includes WikiText2 perplexity and zero-shot downstream task performance and is conducted on LLaMA-7B with 20% sparsity without tuning. The table below shows the results, highlighting that our method significantly reduces WikiText2 perplexity by at least 40%, while maintaining comparable performance in zero-shot downstream tasks compared to LLM-Pruner's various weight importance settings.
>
> |  | WikiText2 ppl | BoolQ | PIQA | HellaSwag | WinoGrande | ARC-e | ARC-c | OBQA | Average |
> | --- | --- | --- | --- | --- | --- | --- | --- | --- | --- |
> | LLM-Pruner (Vector) | 22.28 | 61.44 | 71.71 | 57.27 | 54.22 | 55.77 | 33.96 | 38.40 | 53.25 |
> | LLM-Pruner (Element2) | 19.77 | 59.39 | 75.57 | 65.34 | 61.33 | 59.18 | 37.12 | 39.80 | 56.82 |
> | LLM-Pruner (Element1) | 19.09 | 57.06 | 75.68 | 66.80 | 59.83 | 60.94 | 36.52 | 40.00 | 56.69 |
> | BESA | 11.38 | 66.91 | 72.96 | 61.63 | 58.80 | 62.84 | 34.39 | 35.40 | 56.13 |
>
> If you have any further inquiries or require additional clarification, please don't hesitate to ask.
>
> [A] Jonathan Frankle and Michael Carbin. The lottery ticket hypothesis: Finding sparse, trainable neural networks. In International Conference on Learning Representations, 2019.
>
> [B] Kusupati, Aditya and Ramanujan, Vivek and Somani, Raghav and Wortsman, Mitchell and Jain, Prateek and Kakade, Sham and Farhadi, Ali. Soft Threshold Weight Reparameterization for Learnable Sparsity. Proceedings of the International Conference on Machine Learning, 2020.
>
> [C] Xinyin Ma, Gongfan Fang, and Xinchao Wang. LLM-Pruner: On the Structural Pruning of Large Language Models. Advances in Neural Information Processing Systems, 2023.

---

### Official Review · Reviewer_JpTs · 2023-11-01

**Soundness:** 3 good
**Presentation:** 3 good
**Contribution:** 3 good
**Rating:** 6
**Confidence:** 3

**Summary:**

The authors propose a blockwise model pruning framework for compressing LLMs, which searches for optimal pruning rates for each layer in a differentiable manner. The authors have done experiments on different models and datasets.

**Strengths:**

1. The authors demonstrate the practical speedup of the pruned model in a hardware simulator.
2. The method is parameter-efficient and easy to optimize.
3. The authors have done extensive experiments on language modeling and few-shot learning benchmark datasets. The authors also explore models with different numbers of parameters from 7b to 70b.

**Weaknesses:**

1. It would be better to involve the computational cost of attention weight in Table 4.
2. It would be better to have an ablation study of block-wise pruning. Maybe directly pruning all the models instead of layer by layer.
3. There has been a lot of research on pruning in CV and NLP. The baselines are far from complete, such as "The Lottery Ticket Hypothesis" and its following works. While considering not much work on LLM, I would not penalize too much on it.

Overall, I think the experiments are solid. The novelty is ok, but it would be better to explore more classical pruning methods.

**Questions:**

The model seems not significantly better than SparseGPT with a smaller number of parameters. Could you have significant test?

---

> ### Author Response · Authors · 2023-11-20
> **Response to Reviewer JpTs (part 1/3)**
>
> Thank you for your thoughtful feedback. We appreciate your engagement with our rebuttal and the points you've raised. We have carefully considered each of your concerns and addressed them as follows:
>
> **Q1:** "It would be better to involve the computational cost of attention weight in Table 4."
>
> **A1:** We acknowledge the importance of providing information on the computational cost of running our pruned model in ViTCoD. In response to your suggestion, we have revised Table 4 to include the FLOPs (floating-point operations) of the pruned model. The updated table now offers a more comprehensive view of the efficiency gains achieved by our method. Please refer to the revised table below.
>
> | Layer name | q_proj | k_proj | v_proj | o_proj | gate_proj | up_proj | down_proj |
> | --- | --- | --- | --- | --- | --- | --- | --- |
> | Dense Runtime | 4096 | 4096 | 4096 | 4096 | 10128 | 10128 | 10128 |
> | Average Sparse Runtime | 2232.31 | 2230.50 | 2720.59 | 2698.53 | 5207.53 | 5125.0 | 6850.03 |
> | Dense mFLOPs | 16.78 | 16.78 | 16.78 | 16.78 | 45.09 | 45.09 | 45.09 |
> | Average Sparse mFLOPs | 8.55 | 8.27 | 9.29 | 10.88 | 37.36 | 35.34 | 27.63 |
> | Speedup (Runtime) | 1.83x | 1.84x | 1.51x | 1.52x | 1.94x | 1.98x | 1.48x |
>
> Table 4: Runtime (cycles), computational cost (FLOPs), and speedup across various layers in LLaMA-7B. The term "FLOPs" refers to floating-point operations while "cycles" denotes the number of instruction cycles necessary for the ViTCoD accelerator to perform the associated computational workloads.
>
> **Q2:** "It would be better to have an ablation study of block-wise pruning. Maybe directly pruning all the models instead of layer by layer."
>
> **A2:** Good suggestion. Your insightful question highlights the potential for valuable insights by examining the direct pruning performance across all models. Regrettably, conducting such an experiment proves challenging due to computational limitations, even with our exploration of state-of-the-art strategies such as [microsoft's DeepSpeed](https://github.com/microsoft/DeepSpeed), the int-8 training scheme, and the int8 optimizer provided by the [bitsandbytes library](https://github.com/TimDettmers/bitsandbytes/tree/main). We believe that directly pruning all the models instead of layer by layer could lead to better performance, which can be derived from the experiments in Table 6 and Figure 5 of the Appendix. Table 6 shows that performing our BESA with a larger learning granularity leads to better performance, which is led by the lower quantization error when pruning the model with  a larger learning granularity. Hence,  directly pruning all the models would result in a lower quantization error and thus better performance.

---

> ### Author Response · Authors · 2023-11-20
> **Response to Reviewer JpTs (part 2/3)**
>
> **Q3:** "There has been a lot of research on pruning in CV and NLP. The baselines ..."
>
> **A3:** Thanks for the suggestion. In response, we have conducted experiments using three additional pruning techniques commonly employed in computer vision and natural language processing. These methods, namely iterative pruning (proposed in the Lottery Ticket Hypothesis [A]), threshold pruning [B], and direct mask learning, have been implemented in a blockwise manner for a fair comparison. For iterative pruning, we employ a step-wise approach where we iteratively prune the LLM block with a step of 10%. During each iteration, we update the weights with the reconstruction loss. The results are reported in the table under "Iterative Prune". Then for threshold pruning, we learn a threshold value for each layer within the block while maintaining the block sparsity at 50%. The results are reported in the table under "Threshold Prune". For direct mask learning, we directly learn a sparse mask for the LLM block. The results are reported in the table under "Direct Mask Learning". We opt for iterative pruning as it stands out as one of the most representative methods for pruning, incorporating weight updating. Additionally, we select threshold pruning for its cutting-edge capabilities, showcasing state-of-the-art performance in computer vision models. Lastly, we choose direct mask learning as it represents the most straightforward and directly differentiable pruning method. Notably, the original “Iterative prune” and “Threshold prune” methods use the weight value as a measure of importance. In contrast, we provide the versions that use the weight or Wanda importance metric.
>
> The perplexity results on WikiText2, C4, and PTB datasets are presented below:
>
> |  | BESA | Iterative prune (Wanda) | Iterative prune (Weight) | Threshold prune (Wanda) | Threshold prune (Weight) | Direct mask learning |
> | --- | --- | --- | --- | --- | --- | --- |
> | WikiText2 ppl | 6.86 | 6.88 | 123.48 | 2984.61 | 20351.98 | 1457.53 |
> | C4 ppl | 8.96 | 9.08 | 102.11 | 2191.59 | 21203.34 | 371.47 |
> | PTB ppl | 66.96 | 57.71 | 3929.01 | 4600.38 | 32474.87 | 3927.83 |
>
> These results underscore the unique advantages of our method, particularly its ability to achieve substantial reductions in perplexity. We believe that this thorough exploration of differentiable methods provides a comprehensive understanding of the strengths and capabilities of BESA compared to other baselines.
>
> [A] Jonathan Frankle and Michael Carbin. The lottery ticket hypothesis: Finding sparse, trainable neural networks. In International Conference on Learning Representations, 2019.
>
> [B] Kusupati, Aditya and Ramanujan, Vivek and Somani, Raghav and Wortsman, Mitchell and Jain, Prateek and Kakade, Sham and Farhadi, Ali. Soft Threshold Weight Reparameterization for Learnable Sparsity. Proceedings of the International Conference on Machine Learning, 2020.

---

> > ### Author Response · Authors · 2023-11-21
> > **Response to Reviewer JpTs (part 3/3)**
> >
> > **Q4:** "The model seems not significantly better than SparseGPT with a smaller number of parameters. Could you have significant test?"
> >
> > **A4:** Thank you for your inquiry. After checking, we find that the zero-shot task performance between BESA and SparseGPT is really similar, especially in LLaMA-7B. Therefore, we have performed a thorough analysis of the significance of the performance difference between our proposed method (BESA) and SparseGPT. By conducting experiments with different random seeds (0, 1, 2, 3, and 4), we aimed to ensure the robustness of our findings. The results are shown below and it consistently shows that BESA outperforms SparseGPT across various zero-shot tasks.
> >
> > | Methods | PIQA | BoolQ | HellaSwag | Winogrande | ARC-e | ARC-c | Average |
> > | --- | --- | --- | --- | --- | --- | --- | --- |
> > | Overall Results |  |  |  |  |  |  |  |
> > | BESA | 76.75 +- 0.14 | 72.90 +- 0.66 | 54.37 +- 0.15 | 68.05 +- 0.54 | 70.95 +- 0.15 | 37.75 +- 0.38 | 63.46 +- 0.21 |
> > | SparseGPT | 75.96 +- 0.24 | 73.15 +- 0.88 | 51.28 +- 0.15 | 68.34 +- 1.11 | 70.30 +- 0.68 | 36.81 +- 0.44 | 62.64 +- 0.44 |
> > | Detailed Results |  |  |  |  |  |  |  |
> > | BESA (seed 0) | 76.66 | 72.17 | 54.31 | 67.64 | 70.79 | 37.20 | 63.13 |
> > | SparseGPT (seed 0) | 76.39 | 72.97 | 51.41 | 69.38 | 71.30 | 37.29 | 63.12 |
> > | BESA (seed 1) | 76.55 | 72.63 | 54.45 | 68.82 | 71.13 | 37.54 | 63.52 |
> > | SparseGPT (seed 1) | 75.84 | 73.03 | 51.47 | 67.09 | 69.91 | 36.95 | 62.38 |
> > | BESA (seed 2) | 76.77 | 73.15 | 54.25 | 68.35 | 71.09 | 38.14 | 63.63 |
> > | SparseGPT (seed 2) | 75.84 | 72.23 | 51.19 | 67.64 | 69.53 | 36.35 | 62.13 |
> > | BESA (seed 3) | 76.88 | 73.91 | 54.60 | 67.48 | 70.92 | 37.97 | 63.63 |
> > | SparseGPT (seed 3) | 75.84 | 72.91 | 51.20 | 67.96 | 70.62 | 36.35 | 62.48 |
> > | BESA (seed 4) | 76.88 | 72.63 | 54.24 | 67.96 | 70.83 | 37.88 | 63.40 |
> > | SparseGPT (seed 4) | 75.90 | 74.62 | 51.14 | 69.61 | 70.16 | 37.12 | 63.09 |
> >
> > Table 1: LLaMA-7B accuracies for zero-shot tasks with unstructured 50% sparsity and random seed 0, 1, 2, 3, 4. In the overall results, we present the mean and standard deviation for each item, while the detailed results provide a comprehensive breakdown for every seed, encompassing the entire set of results.

---

### Official Review · Reviewer_qTC9 · 2023-11-01

**Soundness:** 3 good
**Presentation:** 3 good
**Contribution:** 3 good
**Rating:** 6
**Confidence:** 3

**Summary:**

The paper presents a novel pruning technique called Blockwise Parameter-Efficient Sparsity Allocation (BESA) for compressing large language models (LLMs). BESA aims to address the computational footprint and memory consumption issues associated with LLMs by optimizing pruning rates across different layers in a differentiable manner. The proposed method achieves state-of-the-art performance in pruning various LLMs, such as LLaMA1 and LLaMA2, and efficiently prunes them on a single A100 GPU.

**Strengths:**

- BESA is the first differentiable pruning algorithm for LLMs, which allows for efficient optimization of pruning rates across layers.
- The method is parameter-efficient and easy to optimize, exhibiting high efficiency and effectiveness in pruning various LLMs.
- BESA achieves state-of-the-art performance in pruning various LLMs, such as LLaMA1 and LLaMA2, with reduced performance degradation after pruning.
- The proposed method can be jointly optimized with weight-only quantization techniques, further enhancing the compression ratio and efficiency of LLMs.

**Weaknesses:**

- The paper does not provide a detailed analysis of the trade-offs between different pruning rates and their impact on model performance, which could be useful for understanding the optimal pruning strategy.
- The paper does not provide a comprehensive comparison of BESA with other pruning techniques, such as structured pruning, which could help in understanding the relative strengths and weaknesses of the proposed method.

**Questions:**

NA

---

> ### Author Response · Authors · 2023-11-20
> **Response to Reviewer qTC9**
>
> Thank you for your thoughtful feedback. We appreciate your engagement with our rebuttal and the points you've raised. We have carefully considered each of your concerns and addressed them as follows:
>
> **Q1:** "The paper does not provide a detailed analysis of the trade-offs between different pruning rates and their impact on model performance, ... "
>
> **A1:** We apologize for any confusion caused by the placement of ablation studies in our paper. On page 7, in Section 4.2, we discussed and conducted experiments with varying sparsities, which are visually represented in Fig. 3. However, we acknowledge that the figure is presented on page 12 due to the limit of space. From Fig.3, we can see that BESA consistently outperforms other baselines such as SparseGPT and Wanda. The gap is particularly pronounced at large sparsities.
>
> **Q2:** "The paper does not provide a ... structured pruning ... of the proposed method."
>
> **A2:** Your suggestion to include a comparison with structured pruning methods is valid, and we appreciate the insight. Therefore, we conducted experiments about comparing our method with LLM-Pruner [A], the latest structured pruning algorithm we found. After checking the experiments shown in LLM-Pruner, we observed a dramatic decrease in both WikiText2 perplexity and zero-shot performance of some downstream tasks with structured pruning. For a fair comparison, we reimplemented our method in the manner of structured pruning. Specifically, we block-wise pruned the LLM model in the dimension of the attention module's head and the MLP module's hidden dimension. The detailed comparison results are shown in Table 1. The experiment was performed in LLaMA-7B with a sparsity of 20% without tuning. From the table below, it is evident that our method can reduce WikiText2 perplexity by at least 40% while achieving comparable performance in zero-shot downstream tasks compared to LLM-Pruner's different weight importance settings.
>
> | Methods | WikiText2 ppl | BoolQ | PIQA | HellaSwag | WinoGrande | ARC-e | ARC-c | OBQA | Average |
> | --- | --- | --- | --- | --- | --- | --- | --- | --- | --- |
> | LLM-Pruner (Vector) | 22.28 | 61.44 | 71.71 | 57.27 | 54.22 | 55.77 | 33.96 | 38.40 | 53.25 |
> | LLM-Pruner (Element2) | 19.77 | 59.39 | 75.57 | 65.34 | 61.33 | 59.18 | 37.12 | 39.80 | 56.82 |
> | LLM-Pruner (Element1) | 19.09 | 57.06 | 75.68 | 66.80 | 59.83 | 60.94 | 36.52 | 40.00 | 56.69 |
> | BESA | 11.38 | 66.91 | 72.96 | 61.63 | 58.80 | 62.84 | 34.39 | 35.40 | 56.13 |
>
> Table 1: Evaluation of WikiText2 perplexity (ppl, lower is beter) and performance on zero-shot downstream tasks (higher is better) for LLaMA-7B at 20% sparsity without tuning the pruned model.
>
> [A] Xinyin Ma, Gongfan Fang, and Xinchao Wang. LLM-Pruner: On the Structural Pruning of Large Language Models. Advances in Neural Information Processing Systems, 2023.

---

### Official Review · Reviewer_VXVS · 2023-11-05

**Soundness:** 4 excellent
**Presentation:** 3 good
**Contribution:** 4 excellent
**Rating:** 8
**Confidence:** 3

**Summary:**

This paper studies the weight pruning problem of large language models. In order to solve the problems of significant output disturbance and the need for careful hyperparameters tuning in existing layer-wise methods, a novel LLM pruning technique named block-wise parameter-efficient sparsity allocation (BESA) is proposed, with two distinctive characters: minimizing pruning error for individual blocks and ensuring the layer-specific sparsity differentiable. Finally, this paper verified the performance and efficiency of the method through detailed experiments on strong baselines.

**Strengths:**

This paper introduces a novel approach, BESA, for compressing Large Language Models through block-wise pruning with a differentiable sparsity allocation, which maintains the performance of LLMs well and improves computational efficiency compared to existing methods. Besides, this paper is well written, clearly explaining the method of block-wise tuning and parameter-efficient sparsity learning through detailed mathematical and textual expression. Finally, credible experimental design and solid experimental results and increase the credibility of the paper.

**Weaknesses:**

There may be a few things that need to be modified or clarified clearly. In page 3 BLOCK-WISE PRUNING equation (1), it is better to add the meaning of “W” together with “M, F, X, …”; Since the article mentioned that existing methods require meticulous hyperparameter tuning, adding the sensitivity of some vital hyperparameters of the proposed model and clarify the advantage in the appendix will make this paper more convincing.

**Questions:**

1)Considering the abstract mentions existing methods require meticulous hyperparameter tuning, it would be better to study the vital hyperparameter’s sensitivity of the proposed method and experiments are needed, which will make this paper more convincing.
2) Due to the uniqueness of the proposed method BESA, which seeks the optimal pruning rate for each layer, compared to existing methods, will implementing specialized neural network accelerators (ViTCoD in the experiments) consume additional time or make model faster than others?

---

> ### Author Response · Authors · 2023-11-20
> **Response to Reviewer VXVS**
>
> Thank you for your thoughtful feedback. We appreciate your engagement with our rebuttal and the points you've raised. We have carefully considered each of your concerns and addressed them as follows:
>
> **Q1:** "There may be a few things ... add the meaning of “W” together with “M, F, X, …”;"
>
> **A1:** Sorry for any confusion caused by our paper. To clarify, "W" represents all linear weights in a transformer block, "M" refers to the learnable binary mask of these weights, "X" means the input tokens, and "F" is the mapping function that gets the forward result of given "W" and "X".
>
> **Q2:** "Considering the abstract mentions existing methods ... this paper more convincing."
>
> **A2:** Thank you for pointing it out. We provide a comprehensive ablation study of the hyperparameters used in our method on page 12, Appendix Sec. A, which is inclusive of the size of the calibration set, epochs, the step of sparsity candidates, learning granularity, and importance metric.
>
> **Q3:** "Due to the uniqueness of the proposed method BESA, ... make model faster than others?"
>
> **A3:** Good question. Comparing the runtime of the model pruned by our method and other methods in the ViTCoD accelerator can make our experiments more reasonable and convincing. An updated version of Table 4 in our original paper is shown below.
>
> | Layer name | q_proj | k_proj | v_proj | o_proj | gate_proj | up_proj | down_proj |
> | --- | --- | --- | --- | --- | --- | --- | --- |
> | Dense Runtime | 4096 | 4096 | 4096 | 4096 | 10128 | 10128 | 10128 |
> | Average Runtime (SparseGPT) | 2952.22 | 2932.0 | 3041.31 | 2950.13 | 7941.88 | 7865.81 | 7441.44 |
> | Average Runtime (Wanda) | 2887.84 | 2871.34 | 3000.91 | 2461.59 | 7701.41 | 7670.84 | 7388.97 |
> | Average Runtime (BESA) | 2232.31 | 2230.50 | 2720.59 | 2698.53 | 5207.53 | 5125.0 | 6850.03 |
> | BESA Speedup | 1.83x | 1.84x | 1.51x | 1.52x | 1.94x | 1.98x | 1.48x |
>
> Table 4: Runtime (cycles) and speedup across various layers in LLaMA-7B. The term "cycles" denotes the number of instruction cycles necessary for the ViTCoD accelerator to perform the associated computational workloads.
>
>
> Based on the results shown in the previous table, although our method shares the same sparsity with other  methods, we obtain better performance in the ViTCoD accelerator because of our nonuniform sparsity within the block. As shown in Appendix Sec.B, ViTCoD incorporates a dense engine and a sparse engine to perform computation simultaneously and assign the computation workloads to different engines with respect to their sparsity, the model pruned by our methods can have a more balanced workload assignment.

---

### Author Response · Authors · 2023-11-20
**General Response**

Dear Reviewers,

We would like to sincerely thank you all for your thoughtful engagement with our work. Your knowledgeable feedback and probing questions have enormously strengthened this paper and helped guide us toward a more rigorous study. We found the review process to be exceedingly productive and are grateful for the time and expertise you brought.

In response to the diverse and compelling points raised, we have undertaken several major revisions.

- First, we expanded our experiments significantly by implementing additional differentiable pruning techniques like iterative pruning, threshold pruning, and direct mask learning.

- Second, we included more detailed analyses on critical topics like hyperparameter sensitivity, computational cost, and the statistical significance of our results. This level of comprehensive exploration and information will certainly benefit readers.

- Third, upon suggestion, we conducted important new structured pruning experiments comparing against the state-of-the-art algorithm. The findings substantiate our method's ability to dramatically reduce perplexity while preserving capabilities.

- Finally, minor clarifications and reorganizations have enhanced the overall clarity and flow of ideas throughout the paper.

Thank you all once more for your expert perspective and dedication to advancing the field. The code for structured pruning and the additional differentiable pruning techniques will be organized and provided in our Github repo as soon as possible. Please don't hesitate if any part requires further elaboration.

Warmest regards,

Authors

---

### Meta-Review · Area_Chair_u6Nx · 2023-12-06

**Metareview:**

This paper proposes an approach for learning to prune the weights of an LLM via a differentiable pruning operation. Experiments are performed on LLaMA models, where it is found to outperform other pruning methods (SparseGPT/Wanda).

The main strength of this approach is that the method, while not too novel (there has been a long body of work on learning to prune via differentiable masks), is sensible and results in some empirical gains. While I was disappointed by the fact that this approach was doing purely unstructured pruning in the initial submission (hence their speed-up numbers in Table 4 were purely theoretical), during the rebuttal period the authors show that their approach works in the structured pruning case as well, which substantially increases the practical relevance of the approach.

I will add that there has been some work on learning to prune with with targeted pruning rates(albeit at a smaller scale). See, e.g., https://arxiv.org/abs/1910.04732 and https://arxiv.org/abs/2204.00408. However, this work does not seem to engage with this literature.

**Justification For Why Not Higher Score:**

The paper improves over baselines in both unstructured and structured pruning cases.

**Justification For Why Not Lower Score:**

This paper results in only marginal improvements, and the methodological contribution is light.

---

### Decision · Program_Chairs · 2024-01-16

Accept (poster)